# One-Stage Coverage of Leg Region Defects with STSG Combined with VAC Dressing Improves Early Patient Mobilisation and Graft Take: A Comparative Study

**DOI:** 10.3390/jcm11123305

**Published:** 2022-06-09

**Authors:** Gianluca Sapino, Loise Lanz, Aurore Roesti, David Guillier, Sebastien Deglise, Giorgio De Santis, Wassim Raffoul, Pietro di Summa

**Affiliations:** 1Department of Plastic and Hand Surgery, Centre Hospitalier Universitaire Vaudois (CHUV), Rue du Bugnon 46, 1011 Lausanne, Switzerland; gianluca.sapino@chuv.ch (G.S.); loise.lanz@unil.ch (L.L.); wassim.raffoul@chuv.ch (W.R.); 2Department of Vascular Surgery, Centre Hospitalier Universitaire Vaudois (CHUV), Rue du Bugnon 46, 1011 Lausanne, Switzerland; aurore.roesti@chuv.ch (A.R.); sebastien.deglise@chuv.ch (S.D.); 3Department of Plastic Reconstructive and Hand Surgery, Department of Oral and Maxillofacial Surgery—University Hospital, Boulevard de Lattre de Tassigny, 21000 Dijon, France; docteurguillierdavid@gmail.com; 4Department of Plastic and Reconstructive Surgery, University Hospital of Modena, Largo del Pozzo 71, 41100 Modena, Italy; desantis.giorgio@unimore.it

**Keywords:** wound healing, leg ulcer, skin graft, vac therapy

## Abstract

Lower limb skin defects are very common and can result from a wide range of aetiologies. Split thickness skin graft (STSG) is a widely used method to address these problems. The role of postoperative dressing is primary as it permits one to apply a uniform pressure over the grafted area and promote adherence. Focusing on lower limb reconstruction, our clinical study compares the application of V.A.C. (Vacuum Assisted Closure) Therapy vs. conventional dressing in the immediate postoperative period following skin grafting. We included in the study all patients who received skin grafts on the leg region between January 2015 and December 2018, despite the aetiology of the defect. Only reconstructions with complete preoperative and postoperative follow-up data were included in the study. Patients were divided into two groups depending on if they received a traditional compressive dressing or a VAC dressing in the immediate postoperative period. We could retain 92 patients, 23 in the No VAC group and 69 in the VAC group. The patients included in the VAC group showed a statistically significant higher rate of graft take together with a lower immobilisation time (*p* < 0.05). Moreover, a lower rate of postoperative infection was recorded in the VAC group. This study represents the largest in the literature to report in detail surgical outcomes comparing the use of VAC therapy vs. conventional dressing after STSG in the postoperative management of lower limb reconstruction using skin grafts. VAC therapy was used to secure the grafts in the leg region, increasing the early graft take rate while at the same time improving patient mobilisation.

## 1. Introduction

Lower limb skin defects are very common in plastic surgeons’ practice and can result from a wide range of aetiologies including the so-called typical ulcers (venous insufficiency, arterial insufficiency, neuropathy, diabetes and external pressure on body prominences) and the atypical ulcers (caused by inflammatory, neoplastic, vasculopathic, hematological, infectious and drug-induced aetiologies) in which the diagnosis may be difficult and often delayed [1].

These conditions represent a high cost for both the healthcare system and the patients, as they usually require complex and time-consuming medical care, impacting significantly on the quality of life and daily life activities [2].

Restoring skin integrity is of paramount importance in order to prevent deep tissue infection, reduce the fluid leakage and minimise cosmetic impairment.

Split thickness skin graft (STSG) is a widely used method to address this problem: it is on the lower stage of the reconstructive ladder, as it is easy to perform, with a low morbidity and excellent outcomes if the grafted wound bed is appropriate. Indeed, a well-vascularised bed with no intercurrent infection is a fundamental prerequisite in order to maximise the chance for the graft to “take” [3].

Graft survival within the first 48 h is related to the diffusion process of nutrition from the recipient site, while the proper revascularisation process occurs between day 3 and 5. It is during this first period that STSG needs to be immobilised to the wound bed to prevent shearing and hematoma/seroma formation that would disrupt the fragile newly formed vessels network, ultimately determining graft necrosis and reconstruction failure [4].

For this reason, particularly when a skin graft is used to reconstruct the lower limb, patient mobilisation in the early postoperative period is generally avoided in order not to negatively influence graft survival [5].

Besides the surgical technique, the role of postoperative dressing is primary, as it permits one to apply a uniform pressure over the grafted area and promote adherence. Traditionally, compressive, non-adherent and adsorbent dressings are used [6,7].

The Vacuum Assisted Closure device (VAC, KCI Medical s.r.l., Austin, TX, USA) is a modified dressing composed of a reticulated open cell foam covered with occlusive dressing and connected through a tube to a software-controlled therapy unit that creates negative pressure. This device has encountered enormous success in surgical clinical practice for its capacity to promote granulation, reduce bacterial proliferation and improve regional vascularisation. More recently, the use of negative pressure wound therapy (NPWT) has been described as a valid alternative to conventional therapy dressing as it promotes skin graft adherence while removing exudates and oedema from the wound bed [7].

Clinical indications for the use of NPTW are widening through the years and include, among others: open fractures with soft tissue defects, infected and chronic wounds, incisions at risk of breakdown and skin graft [8,9].

Focusing on lower limb reconstruction, our clinical study compares the application of V.A.C. Therapy vs. conventional dressing in the immediate postoperative period following skin grafting. The aim of this study is to evaluate whether V.A.C. Therapy may have a beneficial effect on graft take and may potentially enhance recovery after surgery, especially when considering early patients’ mobilisation. The surgical and functional outcomes (immediate and long-term) are discussed and critically analysed.

## 2. Patients and Methods

A retrospective investigation was performed on a double-centre prospectively maintained database including all patients who received skin grafts between January 2015 and December 2018. Out of the total number of patients, only leg region (therefore excluding the knee and foot) reconstructions with complete preoperative and postoperative follow-up data were included in the study. Further, burnt patients, patients in whom an acellular dermal matrix was used as well as those who underwent a combined procedure (e.g., flap + skin graft) were excluded from the studies to exclude biases. Finally, only patients who signed both an operative consent form and research study consent approval were included in the data analysis.

Patients were divided into two groups depending on whether they received a traditional compressive dressing or a VAC dressing in the immediate postoperative period (group A, skin graft without VAC; group B, skin graft and VAC).

Being a retrospective study, patients were not randomised, and the choice regarding the skin graft dressing to be used was related to the plastic surgeon’s preference. All surgeries were performed by consultant plastic surgeons or under their direct supervision. Patients were stratified by age, BMI (body mass index) and relevant comorbidities (e.g., smoke, diabetes, arteriopathies, infection at the recipient site). The size of STSG, the exposed tissue to be covered and further relevant surgical details were inferred from the operative protocols. All other relevant patient data were gathered from clinical and anaesthesiologic charts from hospital electronic data systems.

Among the outcomes, we evaluated the graft outcome (% take) infection rate, time to healing, hospital stay and the postoperative mobilisation protocol. The graft outcome was assessed at 2 weeks by a plastic surgery resident blinded to the study and was expressed as the percentage of epithelialisation (graft “take”) recorded by clinical inspection.

Cases with >75% graft take were defined as healed, >50% but <75% graft take were considered partially healed (minor complication), while a nonhealing graft/failure was recorded in the case of <50% graft take or when secondary surgical procedures were needed in order to obtain the complete healing of the wound. Surgical complications were recorded according to previous literature [9]. Infection at the surgical site before and after the graft procedure was evaluated using microbiological samples and biopsy.

The study was conducted in accordance with the Declaration of Helsinki (as revised in 2013). Individual consent for this retrospective analysis, including approval for photographic\video documentation, was gathered from all patients. The local ethical committees of the involved institutions approved the study.

### Surgical Technique

All patients underwent surgical debridement until satisfactory bleeding was achieved. Multiple washes with antimicrobial solution (povidone iodine) of the wound bed and its margin were followed by saline washes before graft application, in order to reduce the bacterial contamination of the wound and remove necrotic tissue remnants. After deep bacteriological sampling, antibiotic prophylaxis was administrated in all patients.

After appropriate measurements, the donor area (generally on the external side of the thigh) was infiltrated with 1:500 adrenaline solution. The STSG was harvested using an electric dermatome, maintaining a graft thickness of 0.2 mm. All grafts were meshed at 1:1.5. Skin grafts were stapled to the recipient site after accurate hemostasis.

When the NPWT was used, the VAC therapy system was placed in a sterile condition, as previously described [10].

A nonadherent dressing layer (Adaptic, KCI Medical, Austin, TX, USA) was used as the interface between the foam and the skin graft, to prevent graft tearing when removing the foam. Once the occlusive dressing was applied, a continuous negative pressure therapy at −125 mmHg was set.

On the other hand, when a conventional dressing was applied, sterile cotton gauzes were initially humidified with saline solution, then drained and shaped according to the defect, and finally fixed in a tie-over fashion and further compressed with an external bandage.

The dressing was left intact for 5 days. Venous thrombosis prophylaxis was administrated in all patients. Postoperative orders included limb elevation when at rest and mobilisation according to the surgeon’s preference and patient’s capabilities (e.g., pain). Mobilisation was re-evaluated after the first graft check and depended on the clinical evolution.

In all cases, the donor site area was treated with conventional dressings using paraffin gauzes and was left to secondary healing.

## 3. Statistical Analysis

Continuous variables were compared using independent two-sided t-tests for means and Mann−Whitney U tests for medians—depending on the normality of their distribution, verified using the Shapiro−Wilk test. Contingency tables (e.g., complications and infections) were analysed using a two-sided Chi-square or Fisher’s exact test, as appropriate. Statistical significance was set at a *p*-value < 0.05. Statistical analysis was performed using GraphPad Prism (version 9.0, GraphPad software, La Jolla, CA, USA).

## 4. Results

Over 400 patients were globally identified as receiving STSG in the examined period in the prospectively maintained database. After application of inclusion criteria, we could retain 92 patients, 23 in the No VAC group and 69 in the VAC group.

Patients’ data and wound characteristics are presented in Table 1. Patients of the VAC group were significantly older compared to those of the No VAC group (74 y.o. vs. 61 y.o., *p* < 0.05%, respectively). The two groups were superimposable in terms of BMI and comorbidities. The aetiology of the defects was represented mainly by vascular ulcers (47%) followed by traumatic ulcers (44%) and tumour-related wounds (7%) and were similarly distributed among the two study groups (Figure 1 and Figure 2). The grafted surface was significantly bigger in the VAC group (55 ± 7 cm^2^ vs. 24 ± 6 cm^2^ in the No VAC group, *p* < 0.001) (Figure 3).

Comparing the two groups in terms of graft take, the No VAC group presented a graft take rate of 72% ± 8 (mean ± SEM), while in the VAC group it was 92% ± 2, showing a statistically significant difference (*p* < 0.05).

In the No VAC group, 13 patients out of 23 (56%) were considered completely healed, while a minor complication was seen in six patients (26%) and a graft failure in four patients (18%). In the VAC group, 84% of patients (58 patients out of 69) reached complete healing, 13% (nine patients) experienced minor complications, and only 3% (two patients) were considered as having a graft failure, again showing a significant difference when comparing the two groups in terms of both complete healing and failure percentages.

The final time to complete healing and hospital stay did not differ significantly between groups, despite showing a faster recovery in the VAC group. Mobilisation resulted in being significantly improved in the VAC group, with patients starting mobilisation at 2.6 ± 3 days vs. 4.4 ± 0.5 in the No VAC group (*p* < 0.01) (Table 2).

In the No VAC group, eight out of 23 patients (34%) had preoperative positive microbiological samples, while 13% of patients (three out of 23) newly developed an infection after the surgery despite initially sterile preoperative samples (Table 3).

Instead, in the VAC group, 51% of patients (41 out of 69) were considered infected before the surgery according to the preoperative microbiologic samples (with a significant positive trend when compared to the No VAC group), with seven of those (17%) still presenting infection after the coverage. No patient with initially preoperative sterile samples resulted in being contaminated after surgery, showing a statistically significant difference compared to the No VAC group (*p* < 0.01).

Some patients presented particularly resistant bacteria that were found in both perioperative debridement and postoperative f-up (one patient in the No VAC group and seven patients in the VAC group).

No side effects were reported due to the VAC therapy, apart from an occasional sleep disturbance due to the noise caused by the pump.

## 5. Discussion

Ulcers at the leg represent a serious physical and psychological limit to a patient’s daily life activity, with a high impact on the healthcare system. The split thickness skin graft represents a frequently used tool in the treatment of loss of substance of the leg, particularly when no deep structures (bones, nerves, tendons) are involved/exposed [4,11].

Nevertheless, a proper surgical technique, which includes appropriate donor site debridement and postoperative dressing, is of paramount importance to enhance the final outcome [12].

An initial adherence to the recipient site is necessary for graft survival; otherwise, plasmatic imbibition and revascularisation will not take place, and the graft will slough [13].

The ideal STSG dressing has three main components: elimination of fluid collection, immobilisation of the graft and stabilisation of the graft on an irregular surface [14]. The use of negative pressure dressing seems to address them all. The efficient and firm fixation of the STSG is secured by the negative pressure generated by the VAC pump, at the same time preventing the accumulation of interstitial fluid and exercising a mechanical cleansing of secretion and micro-organisms. In addition, the sealed-off system acts as a barrier against most exogen bacteria [15].

The use of vacuum-assisted devices has been demonstrated to be beneficial, as the microstrain, which is the mechanical stretching of cells under negative pressure, causes them to rapidly divide and proliferate and results in growth factor production needed during wound healing [16,17,18].

Moreover, the removal of exudate and bacteria promoted by the vacuum negative pressure, together with the improvement in oxygenation observed in previous experimental studies, may play a role in improving the quality of skin graft take [19,20,21].

Patient mobility limitation after leg reconstruction with a skin graft can be particularly morbid, especially for aged patients. Our data show how VAC therapy significantly reduced the immobilisation time after surgery without having a negative impact on the surgical outcome. Conversely, a significant improvement of the graft take was seen in the VAC group (*p* < 0.05), despite patients’ earlier mobilisation after surgery (at day 2 on average vs. day 4 in the No VAC group). These results can be explained by considering the constant pressure applied by the device over the grafted surface, which prevents graft shear despite the constant movements of the leg while walking.

Moreover, a significantly higher number of patients in the No VAC group required further surgeries and could not be considered completely healed after the coverage procedure. Our findings are consistent with those of the literature. According to Scherer et al., the use of NPWD to secure the STSG in trauma patients is linked to a statistically significant decrease in the repeated grafting rate in the same patient. However, in their experience, grafted surfaces in the No VAC group were significantly larger, which could lead to the conclusion that the poorer graft survival was due to more extended wounds. In our series, despite the larger defects covered in the VAC group, the graft take rate was superior, suggesting a role of the NPWD in improving graft adherence and take [22].

Korber et al. compared postoperative VAC therapy and standard gauze dressing in 74 mesh grafts (performed in 54 patients) for lower leg vascular ulcers coverage, describing a complete healing in 92% of patient with the VAC vs. 67% with the dressing. In their experience, graft take was also influenced by age and comorbidities (patients < 70 y.o. had better outcomes) [15].

Other case series also revealed improved take rates under postoperative VAC treatment in acute and chronic wounds as well as unproblematic postoperative patient mobilisation [23,24].

However, in those papers, the low number of patients enrolled, the heterogeneous group of wounds and the lack of a comparative cohort treated with conventional dressings made a direct comparison to our results and patient population difficult. Globally, despite the differences in the various study designs, the benefits resulting from a postoperative VAC therapy seem to be shared among authors.

Efficient debridement was a key step in graft take and the removal of potential bacteria. In both groups, we observed a noticeable reduction of postoperative positive microbiological samples (reduction of 87% of the infection rate in the No VAC group and 83% in the VAC group). Interestingly, while in the No VAC group three patients developed a new infection after coverage, in the VAC group no patients with sterile samples at debridement developed a postoperative contamination. This difference was statistically significant, suggesting a potential for VAC dressing to prevent postoperative infection by keeping the wound sealed and aspirating excess fluids [15].

The limitations of this paper need to be acknowledged: first, the retrospective nature of the study and the lack of randomisation. However, the two groups were homogenous according to comorbidities and the ASA score, while patients in the VAC group were significantly older and had bigger defects to cover. This can be explained by the fact that VAC therapy was used when the STSG engraftment appeared less secured. Despite this negative selection bias, graft intake in the VAC group was significantly better, as stated.

Nevertheless, the use of NPWT following a graft coverage procedure of the lower limb is becoming the standard postoperative dressing in our department, which explains the higher number of patients included in the VAC group.

## 6. Conclusions

To the best of our knowledge, this study represents the largest in the literature to report in detail surgical outcomes comparing the use of VAC therapy vs. conventional dressing after STSG in the postoperative management of lower limb reconstruction using skin grafts.

VAC therapy was used to secure the grafts in the leg region (not only following trauma wounds but also on wounds with vascular aetiologies), increasing the early graft take rate while at the same time improving patient mobilisation.

## Figures and Tables

**Figure 1 jcm-11-03305-f001:**
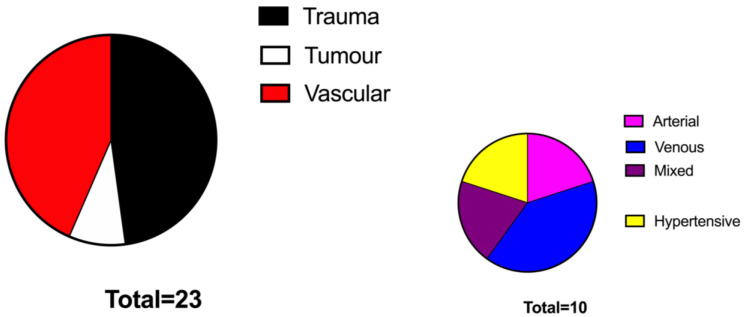
Graphic showing the aetiology of leg wounds in the No VAC group.

**Figure 2 jcm-11-03305-f002:**
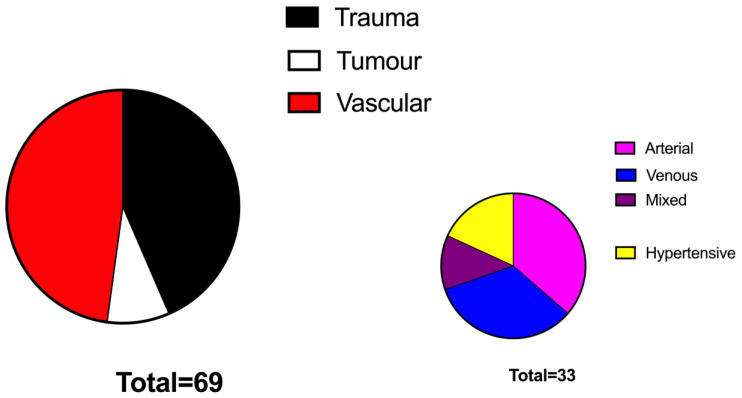
Graphic showing the aetiology of leg wounds in the VAC group.

**Figure 3 jcm-11-03305-f003:**
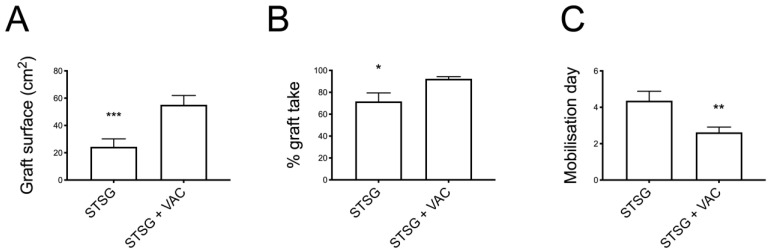
(**A**) Grafted surface was significantly bigger in the VAC group (55 ± 7 cm^2^ vs. 24 ± 6 cm^2^ in the No VAC group, *p* < 0.001, ***); (**B**) In terms of graft take, the No VAC group presented a graft take rate of 72% ± 8 (mean ± SEM), while in the VAC group it was 92% ± 2 (*p* < 0.05, *); (**C**) Mobilisation resulted in being significantly improved in the VAC group, with patients starting mobilisation at 2.6 ± 3 days vs. 4.4 ± 0.5 in the No VAC group (*p* < 0.01, **).

**Table 1 jcm-11-03305-t001:** Patients’ and defects data.

Outcomes	No VAC Group	VAC Group	*p*-Value
No. of Patients	23	69	
Age (years) (mean ± SEM)	61 ± 5	74 ± 3	0.01
Smokers	4	10	>0.05
Diabetes	4	15	>0.05
BMI (mean ± SEM)	25 ± 0.6	26 ± 1.2	>0.05
Grafted surface (cm^2^) (mean ± SEM)	24 ± 6	55 ± 7	0.001

**Table 2 jcm-11-03305-t002:** Surgical outcomes.

Outcomes	No VAC Group	VAC Group	*p*-Value
Graft take (%)	72 (8)	92 (2)	0.01
Grafted surface (cm^2^) (mean ± SEM)	24 ± (6)	55 ± (7)	0.0001
Complete healing	13/23	58/69	0.01
Minor complications	6/23	9/69	>0.05
Graft failure	4/23	2/69	0.01
Hospital stays (days) (mean ± SEM)	11 ± 2	13 ± 1	>0.05
Immobilisation (days) (mean ± SEM)	4 ± 0.5	2 ± 0.3	0.001

**Table 3 jcm-11-03305-t003:** Infection outcomes.

Outcomes	No VAC Group (n.23)	VAC Group (n.69)	*p*-Value
Infection before surgery	8 (34%)	41 (60%)	>0.05
Infection after surgery	4 (17%)	7 (10%)	>0.05
Infection before and after surgery	1 (4%)	7 (10%)	>0.05
Infection only after surgery	3 (13%)	0	0.01

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
