# Peer review of "One-Stage Coverage of Leg Region Defects with STSG Combined with VAC Dressing Improves Early Patient Mobilisation and Graft Take: A Comparative Study"

_jcm, 2022, doi:10.3390/jcm11123305_

Round 1
Reviewer 1 Report
I commend the authors on this excellent report and clinical outcomes. The reduction in wound infection and immobilization days is significant. I would urge the authors to consider a follow-up study that includes economic outcomes in these groups. Well done!
Author Response
We thank the authors for the nice comments. We are glad he could appreciate our efforts and our results.
We are constantly updating our database with the will to improve our results and add other features such as the economic advantage.
Reviewer 2 Report
Introduction
Please define the concept of typical (venous, arterial, pressure and diabetic foot ulcers) and atypical ulcer. (diagnosis and treatment). The standard treatment is based on the principles of the wound bed preparation and wound etiology.Please elaborate on this topic. Which therapies are preferred for the donor site? Which are the others application of NPWT?
Methods
Please define better the inclusion criteria and secondly the exlusion criteria.
How did you treat the donor site? How you measure the area of the wound ?
Which antimicrobial solution did you use? What did you mean for hypertensive wound and tumor related wound (surgical dehiscence?) tumor are are a controindication of NPWT. Infection is defined by clinical and microbiological evalution not only microbiological exam. Did you perform a biopsy for microbiological evaluation?
Discussion
Please increase the discussion with the literature. The references are limited.
Author Response
Introduction
Please define the concept of typical (venous, arterial, pressure and diabetic foot ulcers) and atypical ulcer. (diagnosis and treatment). The standard treatment is based on the principles of the wound bed preparation and wound etiology.Please elaborate on this topic. Which therapies are preferred for the donor site? Which are the others application of NPWT?
We thank the reviewer for the comment. The introduction section has been amended as requested, page 3-4.
Donor site was treated with paraffin gauze such as Jelonetâ and left to secondary healing, page 8
Methods
Please define better the inclusion criteria and secondly the exclusion criteria.
The method section has been amended as requested, page 6.
How did you treat the donor site? How you measure the area of the wound?
Donor site was treated with paraffin gauze such as Jelonetâ and left to secondary healing.
The area of the wound was measured using a ruler on the surgical field following the debridement, multiplying the max length and width of the defect.
Which antimicrobial solution did you use?
The text has been modified, page 7.
What did you mean for hypertensive wound and tumor related wound (surgical dehiscence?) tumor are a contraindication of NPWT.
We thank the reviewer for the comment. The hypertensive ulcer (also known as Martorell ulcer) is a leg ulcer that develop in patients with high blood pressure that is longstanding and often poorly controlled.
Hafner J, Nobbe S, Partsch H, Läuchli S, Mayer D, Amann-Vesti B, Speich R, Schmid C, Burg G, French LE. Martorell hypertensive ischemic leg ulcer: a model of ischemic subcutaneous arteriolosclerosis. Arch Dermatol. 2010 Sep;146(9):961-8. doi: 10.1001/archdermatol.2010.224. PMID: 20855694.
We agree with the reviewer concerning the use of the NPTW on tumor. In our cases we used the NPTW exclusively on skin grafted wound, in some cases following tumor resection, but always after the pathological confirmation of the healthy margins of excision.
Infection is defined by clinical and microbiological evaluation not only microbiological exam. Did you perform a biopsy for microbiological evaluation?
A microbiological biopsy was always performed at the time of surgery. The text has been modified, page 7.
Discussion
Please increase the discussion with the literature. The references are limited.
Thank you for the comment. References have been added as requested.